# Ethanolic Extract from Seed Residues of Sea Buckthorn (*Hippophae rhamnoides* L.) Ameliorates Oxidative Stress Damage and Prevents Apoptosis in Murine Cell and Aging Animal Models

**DOI:** 10.3390/foods12173322

**Published:** 2023-09-04

**Authors:** Zhongjie Hua, Jiachan Zhang, Wenjing Cheng, Changtao Wang, Dan Zhao

**Affiliations:** 1Beijing Key Laboratory of Plant Resource Research and Development, College of Chemistry and Materials Engineering, Beijing Technology and Business University, Beijing 100048, China; 2Institute of Cosmetic Regulatory Science, Beijing Technology and Business University, Beijing 100048, China

**Keywords:** sea buckthorn seed residue, antioxidant, flow cytometric, cell apoptosis, cell cycle, DEPs, GO analysis, KEGG pathway

## Abstract

*Hippophae rhamnoides* L. has been widely used in research and application for almost two decades. While significant progress was achieved in the examination of its fruits and seeds, the exploration and utilization of its by-products have received relatively less attention. This study aims to address this research gap by investigating the effects and underlying mechanisms of sea buckthorn seed residues both in vitro and in vivo. The primary objective of this study is to assess the potential of the hydroalcoholic extract from sea buckthorn seed residues (HYD-SBSR) to prevent cell apoptosis and mitigate oxidative stress damage. To achieve this, an H_2_O_2_-induced B16F10 cell model and a D-galactose-induced mouse model were used. The H_2_O_2_-induced oxidative stress model using B16F10 cells was utilized to evaluate the cellular protective and reparative effects of HYD-SBSR. The results demonstrated the cytoprotective effects of HYD-SBSR, as evidenced by reduced apoptosis rates and enhanced resistance to oxidative stress alongside moderate cell repair properties. Furthermore, this study investigated the impact of HYD-SBSR on antioxidant enzymes and peroxides in mice to elucidate its reparative potential in vivo. The findings revealed that HYD-SBSR exhibited remarkable antioxidant performance, particularly at low concentrations, significantly enhancing antioxidant capacity under oxidative stress conditions. To delve into the mechanisms underlying HYD-SBSR, a comprehensive proteomics analysis was conducted to identify differentially expressed proteins (DEPs). Additionally, a Gene Ontology (GO) analysis and an Encyclopedia of Genes and Genomes (KEGG) pathway cluster analysis were performed to elucidate the functional roles of these DEPs. The outcomes highlighted crucial mechanistic pathways associated with HYD-SBSR, including the PPAR signaling pathway, fat digestion and absorption, glycerophospholipid metabolism, and cholesterol metabolism. The research findings indicated that HYD-SBSR, as a health food supplement, exhibits favorable effects by promoting healthy lipid metabolism, contributing to the sustainable and environmentally friendly production of sea buckthorn and paving the way for future investigations and applications in the field of nutraceutical and pharmaceutical research.

## 1. Introduction

*Hippophae rhamnoides* L., commonly known as sea buckthorn or seaberry, boasts a widespread distribution across Asia, Europe, and North America [1]. It belongs to the *Elaeagnaceae* family and has been traditionally used in a wide range of fields, such as foods, pharmaceuticals, and cosmetics [2]. The diverse components of sea buckthorn offer a plethora of bioactive compounds, making them prized for medicinal and nutritional applications [3,4]. Of noteworthy importance, sea buckthorn seed oil has been subject to extensive investigation due to its exceptional properties, including wound healing, antioxidant, anti-inflammatory, anticancer, antimicrobial, and emollient activities [5,6,7]. Nevertheless, the residues left behind after sea buckthorn seed oil extraction are often discarded or underutilized. However, recent research has highlighted the presence of valuable natural compounds, including flavones, polyphenols, and unsaturated fatty acids, in sea buckthorn seed residues (SBSR) [8,9,10,11].

Despite the promising potential of SBSR for various applications, a comprehensive exploration of its practical utility in raw material development is lacking. Furthermore, the choice of extraction methods can lead to variations in the composition and efficacy of the resulting extracts. Commonly used solvents for SBSR extraction include water and ethanol. Previous studies indicated that aqueous SBSR extracts exhibit hypoglycemic and hypolipidemic effects in type 2 diabetic rats induced with streptozotocin and a high-fat diet [12]. In addition, investigations into different concentrations of organic solvents for sea buckthorn seed extraction have demonstrated varying antioxidative capacities, with ethyl acetate extracts displaying the highest and isopropyl extracts displaying the lowest antioxidative potential [13]. It is also worth noting that SBSR retains a significant amount of liposoluble substances even post-oil recovery. Advanced extraction methods, such as supercritical carbon dioxide, pressurized ethanol, and enzyme-assisted extraction, have been used to isolate valuable components, including tocopherols and monosaccharides, from sea buckthorn pomace and seeds [14].

Numerous in vitro studies have extensively validated the antioxidant capacities of SBSR. These investigations encompass a range of assays, including the estimation of reactive oxygen species generation, measurement of enzymatic/non-enzymatic antioxidant activities/levels, evaluation of peroxisome proliferator-activated receptors levels, and assessment of 3-ethylbenzothiazoline-6-sulfonic acid (ABTS) and 2,2-diphenyl-1-picrylhydrazyl (DPPH) radical scavenging activities [1,7,15]. However, under conditions of oxidative stress, the body initiates a protective stress response to counteract free radicals. This response entails not only direct free radical elimination but also augmentation of antioxidant enzyme activity (e.g., glutathione peroxidase (GSH-Px), superoxide dismutase (SOD), and catalase (CAT)) along with a reduction in the accumulation of peroxidation products like malondialdehyde (MDA) and lipofuscin (LP), thus conferring a protective role [16,17,18]. While existing research on SBSR has predominantly focused on a single level of impact, a comprehensive investigation utilizing animal experiments is needed to elucidate the mechanisms by which HYD-SBSR functions within the body. Therefore, a thorough exploration of the antioxidative mechanism of SBSR is warranted.

In recent years, flow cytometry has undergone significant advancements, enabling the identification of multiple phenotypic subsets, the selection of individual cells, and even cell isolation using sorting. This technology plays a pivotal role in clinical settings by identifying aberrant cells, quantifying excessive or reduced populations of specific cells, and monitoring changes in tracked cell populations [19]. In this study, in conjunction with mouse experiments, the levels of antioxidant enzymes and peroxides were also analyzed in vivo. A proteomics analysis was conducted to scrutinize changes in differentially expressed proteins (DEPs) post-HYD-SBSR administration. Subsequent Gene Ontology (GO) analysis and Encyclopedia of Genes and Genomes (KEGG) pathway enrichment analysis were performed to provide further insights into the impact of DEPs.

B16F10 cells, a melanoma cell line derived from mice, possess high metabolic activity leading to reactive oxygen species (ROS) production, potential stress response activation, rapid growth facilitating timely observations, and relevance to simulating disease-related conditions [20,21]. These attributes collectively position B16F10 cells as a valuable model for investigating oxidative stress and cellular responses. The D-galactose-induced mouse aging model offers simplicity and ease of operation, enabling rapid inter-group comparisons. It has been demonstrated as an effective simulation of aging effects and finds wide application in antioxidant research [22,23].

Prior research has demonstrated that sea buckthorn seed products can contribute to reducing blood glucose and lipid levels in obese mice [12,24]. Using them as supplementary food additives holds the potential to regulate lipid metabolism in the body. This approach could also mitigate certain suboptimal health conditions arising from high-sugar diets. The primary objective of this study was to assess the effects of the hydroalcoholic extract from sea buckthorn (*Hippophae rhamnoides* L.) seed residues (HYD-SBSR) at the cellular and physiological levels while establishing a correlation between in vitro and in vivo experimental findings. This not only supports the broader application of sea buckthorn seed by-products in food but also addresses environmental concerns linked to sea buckthorn production, thereby contributing to sustainable and eco-friendly industrial development.

## 2. Materials and Methods

### 2.1. Preparation of HYD-SBSR

HYD-SBSR was prepared following the methodology outlined in [25]. Ground powder of sea buckthorn seed residues (provided by Qinghai Kompu Biotechnology Co., Ltd., Xining, China, dried, crushed, and sifted) was mixed with an 80% ethanol aqueous solution at a ratio of 8:1 (liquid to solid ratio, mL/g). The mixture was then extracted for 1.5 h. After extraction, centrifugation was carried out at 400× *g* and 4 °C for 15 min. The resulting supernatant was collected, and the extraction process was repeated twice to enhance the yield and purity of the extracted components. The collected supernatants from each extraction were combined to create a consolidated sample. The ethanol in the sample was subsequently removed using rotary evaporation under vacuum conditions at a controlled temperature of 40 °C. This evaporation process facilitated solvent removal, concentration of the extracted substances, and conversion into a fine powder. The collected HYD-SBSR was stored securely for future investigations. The flavonoid, procyanidin, and total phenolic contents reached up to 354.00 ± 21.00 mg RE per g DW, 319.31 ± 11.70 mg CE per g DW, and 271.24 ± 91.30 mg GAE per g DW, respectively [25].

### 2.2. Cell Culture

B16F10 cells (obtained from the Cell Resource Center, Institute of Basic Medicine, Chinese Academy of Medical Sciences) were cultured in Dulbecco’s Modified Eagle Medium (DMEM, Grand Island Biological Company, New York, NY, USA), supplemented with 10% heat-inactivated fetal calf serum(FBS, GIBCO, Darmstadt, Germany), and a 1% antibiotic–antimycin solution consisting of 100 units/mL penicillin /streptomycin and 100 U/mL amphotericin (GIBCO, Darmstadt, Germany) at 37 °C in a humidified incubator (Shanghai Shengke, Shanghai, China) under 5% CO_2_.

### 2.3. Cell Viability

Cell viability was determined using a dimethylthiazole-2-yl)-2,5-diphenyltetrazolium bromide (MTT, Sigma, Welwyn Garden City, UK) assay. B16F10 cells were seeded into a 96-well plate at a concentration of 5000 cells per well at 37 °C and were incubated in a cell incubator with 5% CO_2_ for 12 h. The cells were treated with H_2_O_2_ or HYD-SBSR at different concentrations for a period. Details of the concentration values are provided in Section 2.6 and Section 2.7.

The MTT assay involved the addition of a 100 μL mixture of MTT solution (5 mg/mL) and DMEM at a volumetric ratio of 1:5. This treatment was carried out for 4 h, followed by the addition of 150 μL of DMSO to dissolve the resulting products. After a 10-minute incubation at 37 °C, the absorbance was measured at a wavelength of 490 nm to determine the results. 

Cell viability was calculated following the equation below.

Cell Viability (%) = OD_t_/OD_0_ × 100%
(1)

where OD_t_ represents the experimental group absorbance minus zeroing group absorbance and OD_0_ represents the control group absorbance minus zeroing group absorbance. 

### 2.4. Establishment of the H_2_O_2_-Induced B16F10 Cell Oxidative Stress Model

A model of acute ROS-induced B16F10 cells was established using a 4-hour treatment with hydrogen peroxide(H_2_O_2_).

Different concentrations of H_2_O_2_ ranging from 0.4 to 44.1 mM were utilized for treatment, each lasting for 4 h. At least 6 parallel wells were used for each group. Cell viability was measured. IC_50_ parameters were selected to establish the oxidative stress model, striking a balance between eliciting significant effects within a reasonable timeframe and ensuring reproducibility.

### 2.5. Protective and Repair Effects of HYD-SBSR on the H_2_O_2_-Induced B16F10 Cell Oxidative Stress Model

To explore the protective effects of HYD-SBSR on cells, B16F10 cells seeded in a 96-well plate were treated with 0.05, 0.1, 0.2, and 0.4 mg/mL of HYD-SBSR for 24 h, cleaned twice with PBS, and treated with 8.8 mM H_2_O_2_ for 4 h. Cell viabilities were detected using an MTT assay. 

Additionally, the repair effects of HYD-SBSR were investigated. B16F10 cells were first exposed to H_2_O_2_ for 4 h to establish the oxidative stress model. Subsequently, the cells were treated with different concentrations of HYD-SBSR for 24 h. Cell viability was assessed using an MTT assay.

### 2.6. GSH-Px, CAT, SOD, and MDA in B16F10 Cells

The experimental procedure involved the treatment of B16F10 cells with H_2_O_2_ (8.8 mM) for a duration of 4 h to establish the H_2_O_2_-induced cell oxidative stress model. Subsequently, the cells were subjected to treatment with varying concentrations of HYD-SBSR (0, 0.05, 0.10, 0.20, and 0.40 mg/mL) for a period of 24 h. The group of cells without any HYD-SBSR treatment served as the model group for comparison.

Cells at a density of 1 × 10^5^ were planted in a six-well plate and incubated overnight. Cells were treated with HYD-SBSR (0, 0.05, 0.10, 0.20, and 0.40 mg/mL) for 24 h and then treated with H_2_O_2_ (8.8 mM) for 4 h. Only cells treated with HYD-SBSR were used as the HYD-SBSR control. Cultured cells were washed with PBS. The cells were lysed using 200 µL Western and IP cell lysis buffer (Beyotime, Nanjing, China), followed by a 12,000× *g* centrifugation for 10 min to collect the supernatant. The contents of GSH-Px, CAT, SOD, and MDA were then determined in a 96-well plate (Corning, Corning, NY, USA), according to the instructions of the GSH-Px, CAT, SOD, and MDA detection kit (Beyotime, Nanjing, China). Protein content was measured using a BCA protein kit from Beyotime. The contents of these parameters were calibrated using the protein content and expressed as micrograms per milligram of protein.

### 2.7. Flow Cytometry Analysis for Cell Cycle Distribution and Apoptosis

Annexin V-PE is a fluorescently labeled protein that binds to calcium-dependent phospholipids with a strong affinity for phosphatidylserine (PS) binding sites, similar to Annexin V-FITC. During the early stages of cell apoptosis, the loss of membrane symmetry exposes PS on the outer surface of the cell membrane. In both the early and late stages of apoptosis, PS is present on the outer surface of the cell membrane. Importantly, early apoptotic cells maintain membrane integrity, preventing the entry of 7-AAD. In contrast, late apoptotic cells exhibit compromised membrane integrity and can be co-stained with Annexin V-PE or V-FITC and 7-AAD. This staining method allows for the differentiation and identification of cells at different stages of apoptosis based on variations in phospholipid exposure and membrane integrity [26,27].

In this study, cell death was detected and analyzed with flow cytometry using FACS Calibur (Becton Dickinson Biosciences, San Jose, CA, USA).

Seeding was performed with a cell density of 1.5 × 10^5^ cells per well in a 6-well plate, with 3 wells per group. The model group was initially exposed to 8.8 mM H_2_O_2_ for 4 h, followed by treatment with 0.1 mg/mL HYD-SBSR for 24 h. Conversely, the HYD-SBSR group underwent the opposite treatment sequence. Then, cells from each group were harvested using trypsin digestion for subsequent analyses. After treatment, approximately 10,000 cells were obtained from each sample. Staining of cell samples was performed using Annexin V-PE and 7-AAD (Annexin V-PE Apoptosis Detection Kit I, BD Bioscience, San Jose, CA, USA). Experimental data were obtained and analyzed using CellQuest (Becton Dickinson Immunocytometry Systems, San Jose, CA, USA). Annexin V-PE and 7-AAD fluorescence (Becton, Dickinson, ND, USA) were used for two-parameter point plots. 

Cell populations were separated as follows: viable cells-Annexin V-PE-negative and 7-AAD-negative (Q4, Annexin V-PE-/7-AAD−); early apoptotic cells-Annexin V-PE-positive and 7-AAD-negative (Q3, Annexin V-PE+/7-AAD−); late-apoptotic cells or dead cells- Annexin V-PE-positive and 7-AAD-positive (Q1, Annexin V-PE+/7-AAD+); and cells debris-Annexin V-PE-negative and 7-AAD-positive (Q2, Annexin V-PE-/7-AAD+). The proportion of 4 groups was calculated using FlowJo software (version 10, FlowJo, LLC, Ashland, OR, USA). The total apoptosis rate was calculated by summing the rate of early apoptotic cells and late apoptotic cells.

### 2.8. Animals and Treatment 

A D-galactose-induced aging mice model was established to investigate the protective effect of HYD-SBSR. The experiment was conducted using specific pathogen-free (SPF)-grade Institute of Cancer Research (ICR) male mice (*Mus musculus*).

Fifty male SPF ICR mice with an average body weight of (23 ± 3) g were ordered from Beijing Vital River Laboratory Animal Technology Co., Ltd. (Beijing, China) and acclimated for one week. They were then randomly divided into 5 groups (*n* = 10): control group, model group, HYD-SBSR high-dose group (HYD-SBSR-H), HYD-SBSR medium-dose group (HYD-SBSR-M), and HYD-SBSR low-dose group (HYD-SBSR-L). Each group contained 10 mice.

Except for the control group, the remaining groups were subjected to daily intraperitoneal injections of 10% D-galactose solution (100 mg/kg body weight). Control-group mice received an equivalent volume of 0.9% physiological saline. The low-dose, medium-dose, and high-dose groups were orally administered 100, 300, and 600 mg/(kg body weight) of HYD-SBSR, respectively. The control and model groups were administered an equivalent volume of physiological saline. 

The treatment duration lasted 42 days. Throughout the intragastric administration period, the mice were maintained in an environment with a temperature of 22 to 25 °C, relative humidity of 50% to 60%, and a 12-hour light (08:00–20:00) and 12-hour dark cycle under fluorescent illumination. Bedding was changed every 3 days. All mice received a standard diet and had access to water.

No mice experienced mortality over the course of the entire experiment, and all subjects were included in this study. The Guide for the Care and Use of Laboratory Animals (National Institutes of Health, Stapleton, NY, USA) was strictly followed in designing all animal experimental procedures. Ethical approval for all animal experiment procedures was granted by the Experimental Animal Welfare Ethics Committee of Beijing Experimental Animal Research Center (BLARC-2017-A015).

### 2.9. Collection of Mouse Experimental Samples

Following euthanization with cervical dislocation, segments of liver and brain tissues were frozen in liquid nitrogen first and then homogenized under an ice bath to prepare 10% homogenate in a 1:9 (*w*/*v*) ratio with pre-chilled physiological saline, which was centrifuged at 3 000 r/min and 4 °C for 15 min. The supernatants were removed to determine tissue biochemical indexes (GSH-Px, CAT, SOD, MDA, lipofuscin, and total antioxidant capacity). All indexes were determined following the manufacturer’s instructions.

Additionally, a proteomics assay was conducted on liver samples from the low-dose group, using TMT labeling quantitative proteomics technology (CapitalBio Technology. Beijing, China). The whole process included protein sample preparation, TMT labeling, high-performance liquid chromatography (HPLC) fractionation, liquid chromatography-tandem mass spectrometry (LC-MS/MS) analysis, and proteomics data analysis. A detailed protocol for this analysis is provided in the Appendix A.

### 2.10. Differentially Expressed Proteins (DEPs) and GO Enrichment and KEGG Pathway Enrichment Analyses

The DEPs satisfied the following conditions: average ratio-fold change >1.1 (up-regulation) and <0.9 (down-regulation) and a *p*-value < 0.05.

Functional classification of the DEPs was performed according to the Gene Ontology (GO) annotation and enrichment analysis. The Encyclopedia of Genes and Genomes (KEGG) Orthology-Based Annotation System (KOBAS) v2.0 was used. The enrichment analysis of GO function significance unveiled functional categories that exhibited significant enrichment in the pool of differential proteins in comparison with the broader genomic background. This analysis entailed the submission of all differential proteins to the GO database (http://www.geneontology.org/ Map each term of org/, accessed on 20 July 2023), where the count of proteins for each term was calculated. Subsequently, hypergeometric tests were applied to pinpoint GO entries that exhibited substantial enrichment among the differential proteins, relative to the genome background. After multiple tests and corrections, GO terms with a *p*-value of ≤0.05 were considered significantly enriched in the set of differential proteins. These DEPs were then categorized into three primary classifications, namely, biological process (BP), cell component (CC), and molecular function (MF). The KEGG database was used to identify enriched pathways. A two-tailed Fisher’s exact test was used to evaluate the enrichment of DEPs relative to all the identified proteins within specific pathways. A pathway achieving a corrected *p*-value of < 0.05 was deemed to be significant. These pathways were subsequently classified into hierarchical categories in accordance with the KEGG website’s structure.

### 2.11. Statistical Analysis 

The experimental data was analyzed using SPSS 19.0 (SPSS, Chicago, IL, USA) and GraphPad Prism 9.0 (GraphPad Software, La Jolla, CA, USA) software. Single factor ANOVA analysis was used for comparisons between groups, and a *t*-test was used for pairwise comparisons. A *p* < 0.05 was considered statistically significant, and the results were shown as mean ± standard deviation.

## 3. Results

### 3.1. Establishment of the H_2_O_2_-Induced B16F10 Oxidative Stress Model

The impact of H_2_O_2_ on cell viability and the induction of oxidative stress in B16F10 cells were investigated. An MTT assay was used to assess the viability of B16F10 cells exposed to varying concentrations of H_2_O_2_, ranging from 0.4 to 44.1 mM. The results demonstrated that as the dose of H_2_O_2_ increased, cell viability progressively decreased (Figure 1). Conversely, as the H_2_O_2_ concentration decreased, cell viability exhibited an increase, and the effects on B16F10 cells were nearly 100% when the H_2_O_2_ concentration was below 4.4 mM. However, when the H_2_O_2_ concentration exceeded 17.6 mM, the viability of B16F10 cells was almost reduced to 0%. For the purpose of establishing an oxidative stress model, the IC_50_ value was chosen, which represents the concentration at which cell viability is reduced by 50%. Specifically, the viability of B16F10 cells treated with 8.8 mM H_2_O_2_ was measured to be (50.31 ± 2.53) %. Therefore, the oxidative stress model was established using the conditions of 8.8 mM H_2_O_2_ exposure for a duration of 4 h.

### 3.2. Protective and Repair Effects of HYD-SBSR on H_2_O_2_-Induced B16F10

The investigation delved into the impact of varying concentrations of HYD-SBSR (ranging from 0.05 to 0.4 mg/mL) on the viability of B16F10 cells (Figure 2A). Remarkably, when treated with 0.05 and 0.1 mg/mL of HYD-SBSR, cell viability surpassed the 80% mark, underscoring the compound’s low cytotoxicity and propensity to sustain cell survival rates beyond 80%. Consequently, the concentration of 0.1 mg/mL HYD-SBSR was deemed suitable for subsequent experimental conditions.

Additionally, this study examined the potential of both pre- and post-treatment with HYD-SBSR in the context of H_2_O_2_-induced oxidative stress. The examination encompassed assessments of cell viability (Figure 2A) as well as apoptosis (Figure 2B,C). 

In both of the studies on different HYD-SBSR treatments, the respective model group generally exhibited significant oxidative stress (*p* < 0.01) compared with their respective control group. The pre-treatment with HYD-SBSR at 0.1 mg/mL (HYD-SBSR + 8.8 mM H_2_O_2_ in Figure 2A) resulted in a substantial augmentation of B16F10 cell viability, showcasing a clear contrast with the model group. Conversely, the other concentrations failed to induce similar beneficial effects. Furthermore, the post-treatment strategy using 0.1 mg/mL HYD-SBSR (8.8 mM H_2_O_2_+ HYD-SBSR in Figure 2A) also exhibited the ability to enhance cell viability when juxtaposed with the model group.

Flow cytometry analysis was used to examine cell apoptosis at a concentration of 0.1 mg/mL HYD-SBSR (Figure 2B). This enabled the computation of rates for early, late, and total apoptosis (Figure 2C). Strikingly, the model group demonstrated the highest levels of early, late, and total apoptosis among the various groups studied. Notably, the proportion of cells undergoing early apoptosis eclipsed that of cells undergoing late apoptosis. The pre- and post-treatment regimens with HYD-SBSR were both capable of attenuating the pro-apoptotic effects induced with H_2_O_2_, thereby reducing the number of cells undergoing either early or late apoptosis. Interestingly, among the studied interventions, the pre-treatment with HYD-SBSR exhibited a relatively stronger protective effect against H_2_O_2_-induced oxidative stress.

To gain deeper insights into the protective and repair properties of HYD-SBSR in B16F10 cells, intracellular antioxidant enzyme levels (GSH-Px, SOD, and CAT) and lipid peroxidation products, such as MDA content, were detected. 

As illustrated in Figure 2D, the analysis involved cells subjected to different treatments, including H_2_O_2_ and HYD-SBSR, as well as pre- and post-treatment strategies in H_2_O_2_-induced models. In comparison with the control group, a discernible trend emerged that the GSH-Px, CAT, and SOD levels notably decreased, while the MDA content exhibited a marked increase in the model group (*p* < 0.01). Conversely, cells treated solely with HYD-SBSR demonstrated minimal alterations in these indices (*p* > 0.05), with the exception of CAT (*p* < 0.05), as compared to the control group.

Significantly, the post-treatment with HYD-SBSR following H_2_O_2_ exposure (H_2_O_2_ + HYD-SBSR treatment) yielded a substantial increase in the levels of cellular enzymes (GSH-Px, CAT, and SOD) (*p* < 0.01) coupled with a reduction in MDA contents (*p* < 0.01) when contrasted with the model group. This intervention resulted in the GSH-Px, CAT, SOD, and MDA contents reaching values of (303.06 ± 19.24) mU/mg protein, (107.95 ± 1.29) mU/mg protein, (237.27 ± 21.98) mU/mg protein, and (8.12 ± 0.96) μmol/g protein, respectively, indicative of the potential reparative influence of HYD-SBSR. Similarly, the pre-treatment regimen involving HYD-SBSR (HYD-SBSR + H_2_O_2_ treatment) displayed a comparable trend. However, it was noted that the enhancement of GSH-Px, CAT, and SOD levels was relatively subdued, and the inhibition of MDA was less pronounced compared with the post-treatment strategy using HYD-SBSR.

### 3.3. Antioxidant Enzyme and Peroxide Levels In Vivo

D-galactose-induced aging mice were utilized to investigate antioxidant enzyme and peroxide levels in vivo. The continuous administration of D-galactose led to disturbances in glucose metabolism within mice cells, disrupting the body’s antioxidant defense system and resulting in the accumulation of free radicals. This, in turn, triggered oxidation reactions leading to the production of substances like LP and MDA, culminating in bodily aging.

Apart from assessing the total antioxidant capacity (T-AOC), the activities of key antioxidants, including SOD, GSH-Px, and CAT, offered direct insight into the body’s ability to counteract free radicals. For the analysis, liver and brain tissues were collected from mice to measure these antioxidant enzyme and peroxide levels (Figure 3). In comparison with the control group, the activities of SOD, GSH-Px, CAT, and T-AOC in the liver and brain tissues of the model group exhibited significant reductions (*p* < 0.01), effectively confirming the successful establishment of the D-galactose-induced aging model.

As depicted in Figure 3A, liver GSH-Px activities were significantly elevated in both the HYD-SBSR-L group (77.85 ± 13.10 mU/mg protein) and the HYD-SBSR-H group (48.61 ± 9.12 mU/mg protein) in comparison with the model group (*p* < 0.01). However, no statistically significant differences were observed between the HYD-SBSR-M group and the model group (*p* > 0.05). Conversely, for brain tissue, no noteworthy distinctions were evident between the three doses of HYD-SBSR and the model group (*p* > 0.05). 

Figure 3B depicts the activities of CAT in both liver and brain tissues. Mice in the HYD-SBSR-L group exhibited a marked increase in liver CAT activity (587.66 ± 25.63 U/mg protein, *p* < 0.01), surpassing that in the model group (551.00 ± 41.85 U/mg protein). In contrast, medium and high doses of HYD-SBSR did not significantly affect liver CAT activities when compared to the model group (*p* > 0.05). However, a substantial impact of HYD-SBSR on CAT activity in brain tissue was evident, showing values of 42.42 ± 6.64 U/mg protein as compared with the model group. Notably, all three doses of HYD-SBSR significantly enhanced brain CAT activity (*p* < 0.01) in a dose-dependent manner. 

Turning to liver SOD activities (Figure 3C), both the HYD-SBSR-M and HYD-SBSR-L groups exhibited significantly higher levels compared with the model group (*p* < 0.05). No significant differences were observed in liver SOD activities between HYD-SBSR-H and the model group (*p* > 0.05). Similar patterns emerged for brain SOD activities, showcasing a dose-dependent trend. Medium- and high-dose groups (HYD-SBSR-M and HYD-SBSR-H) displayed significantly elevated brain SOD activities (*p* < 0.01) compared with the model group, whereas no substantial differences were observed between the HYD-SBSR-L and model groups (*p* > 0.05).

The total antioxidant capacities (T-AOC) of both liver and brain tissues in the model group were notably lower than those in the control group (Figure 3F, liver, *p* < 0.01; brain, *p* < 0.05). HYD-SBSR treatment brought about a significant enhancement in T-AOC levels within the liver and brain tissues. T-AOC levels in the HYD-SBSR-L group surpassed those in the model group (liver, *p* < 0.05; brain, *p* < 0.01). However, no statistically significant differences in T-AOC levels were observed between the HYD-SBSR-M and HYD-SBSE-H groups and the model group (*p* > 0.05).

The levels of MDA (Figure 3D) and LP (Figure 3D) in the D-galactose-induced aging model group were significantly higher than those in the control group (LP, *p* < 0.05; MDA, *p* < 0.01). After HYD-SBSR treatment, the levels of LP and MDA in liver and brain were lower than those in the model group (*p* < 0.01).

### 3.4. TMT-Based Quantitative Proteomics Analysis of Liver Tissue

In order to further clarify the mechanism underlying HYD-SBSR in vivo, we conducted a TMT-based quantitative proteomics analysis coupled with bioinformatics assessment. Leveraging the mouse (mmu) protein database, a comprehensive total of 5510 reliable proteins were identified under conditions with less than a 1% false discovery rate (FDR). Figure 4 illustrates the screening process of DEPs using the fold change (FC) value and *p*-value as pivotal criteria (FC ≥ 1.1 or ≤0.9 and *p* ≤ 0.05). In comparison with the control group, the model group exhibited a set of 79 DEPs, among which 32 were upregulated and 47 were downregulated (Appendix A). Similarly, the HYD-SBSR group demonstrated 100 DEPs, consisting of 63 upregulated and 37 downregulated proteins, relative to the model group (Appendix A). Intriguingly, six DEPs were found to overlap between the DEPs for both groups (Figure 4C, Table 1). Among these DEPs, three (Eef1e1, Farp1, and Aga) displayed an intriguing pattern of being upregulated in the model group and concurrently downregulated in the HYD-SBSR group (Table 1).

We performed GO enrichment analysis separately on the DEPs identified in the control versus model group (Figure 5A) and the model versus the HYD-SBSR group (Figure 5B). In the comparison between the control and model groups, the DEPs exerted significant impacts on 11 types of biological processes (BPs), 6 types of cellular components (CCs), and 6 types of molecular functions (MFs) (*p* < 0.05). Among the top five enriched GO terms were processes such as catecholamine metabolic process, positive regulation of cholesterol biosynthetic process, cellular response to glucagon stimulus, amino acid binding, and carboxylyase activity. In the comparison between the model and HYD-SBSR groups, the DEPs significantly influenced 9 BP, 10 CC, and 10 MF categories (*p* < 0.05). Among the top five enriched GO terms were processes like negative regulation of intestinal phytosterol and cholesterol absorption, 3′-phosphoadenosine 5′-phosphosulfate biosynthetic process, sulfate adenylyl transferase (ATP) activity, ATP-binding cassette (ABC) transporter complex, and adenylyl sulfate kinase activity.

Further analysis revealed the enriched KEGG pathways (*p* < 0.05). In the analysis of the control vs. the model group (Figure 6A), the top three pathways were selected including cysteine and methionine metabolism, metabolic pathways, and biosynthesis of amino acids. In the model vs. the HYD-SBSR group (Figure 6B), there were 10 pathways that showed significance (*p* < 0.05), including selenocompound metabolism, PPAR signaling pathway, lysosome, ABC transporters, metabolic pathways, fat digestions and adsorption, glycerophospholipid metabolism, complement and coagulation cascades, cholesterol metabolism, and sulfur metabolism.

## 4. Discussion

H_2_O_2_ is known to readily diffuse into nuclear tissue, leading to the onset of various oxidative stress conditions. Due to this property, exogenous H_2_O_2_ has often been used in studies to induce oxidative stress damage and apoptosis. This approach helps researchers investigate the cellular protective and repair effects of bioactive substances. Cellular damage often results in a decline in the body’s ability to eliminate H_2_O_2_, causing an accumulation of ROS and the initiation of lipid peroxidation reactions. In turn, this leads to the formation of products such as MDA, which can damage vital biological molecules like proteins and lipids. To maintain cellular homeostasis, the body relies on its antioxidant enzyme system, which includes key enzymes like GSH-Px, CAT, and SOD. These enzymes play a crucial role in breaking down hydrogen peroxide generated during metabolism and neutralizing ROS and other free radicals that arise during oxidative stress. In this study, an H_2_O_2_-induced B16F10 model was established to evaluate the protective and repair abilities of HYD-SBSR at the cellular level in vitro. The results indicated that HYD-SBSR exhibited superior protective capabilities, as evidenced by its tendency to enhance cell viability and reduce H_2_O_2_-induced apoptosis (Figure 2A–C). However, the cellular antioxidant assays hinted that while HYD-SBSR demonstrated a protective effect, its repair effect showed even greater promise (Figure 2D). Although the findings did not align completely when comparing the two treatment types, it was evident that HYD-SBSR held the potential to counteract oxidative stress. This aligned with the findings from our laboratory’s analysis of flavonoid, procyanidin, and total phenolic contents present in HYD-SBSR [25].

To address the limited reports available on the impact of HYD-SBSR in vivo, a more in-depth investigation into its potential repair effects on antioxidant enzymes and peroxides in mice was carried out. The administration of three different levels of HYD-SBSR demonstrated the ability to elevate antioxidant enzyme levels in the liver and brain. These results suggested that the liver, known for its detoxification and metabolism functions, played a crucial role in maintaining cellular homeostasis.

To delve further into the molecular pathways underlying the antioxidant effects of HYD-SBSR in vivo, a proteomics approach was used. Importantly, three DEPs, namely, Eef1e1, Farp1, and Aga, were screened, which were upregulated in the model group and decreased after the treatment of HYD-SBSR. Conversely, one DEP (Pigr) demonstrated the opposite expression pattern. 

Of significance among the DEPs is the protein Aga, also referred to as aspartylglucosaminidase, a lysosomal enzyme that starts as an inactive precursor molecule and is swiftly activated within the endoplasmic reticulum [28]. An Aga deficiency leads to aspartylglycosaminuria, a lysosomal disorder causing impaired glycoprotein degradation [29]. Remarkably, research by Ulla Dunder et al. illustrated that a 10% increase in Aga activity resulted in a 20% reduction in aspartylglycosaminuria accumulation [30]. The increased expression of Aga was detected in the model group, probably indicating a self-regulation of the body against disordered glucose metabolism.

Eef1e1, also known as eukaryotic translation elongation factor 1 ε 1, plays a role in protein synthesis and cell differentiation [31,32]. It positively modulates the ATM response to DNA damage [33] and can be induced by DNA-damaging agents like UV, Adriamycin, actinomycin D, and cisplatin. The increased expression of Eef1e1 in the model group is understandable, given its connection to the DNA damage response. The decrease in Eef1e1 protein levels after HYD-SBSR treatment suggested a potential reparative effect.

Moreover, Farp1 was identified as one of the guanine nucleotide exchange factors, which belongs to a family of regulatory proteins for Rho GTPases, influencing various cellular processes [34]. Farp1 interaction with cell surface proteins regulates neuronal development [35,36], and high expression is associated with lymphatic invasion and metastasis [37]. In the context of D-galactose-induced aging mice with disrupted glucose metabolism and ROS accumulation, upregulated Farp1 likely combats oxidative stress and inflammation in the model group. Conversely, the decreased Farp1 expression upon HYD-SBSR administration implies a beneficial effect. 

Pigr (polymeric immunoglobulin receptor precursor) is a single transmembrane protein. Its expression was subsequently upregulated after HYD-SBSR treatment. The impact on the MEK/ERK pathway suggests potential attenuation of liver injury in mice [38].

Regarding the GO and KEGG pathway analyses, our findings revealed that HYD-SBSR significantly influences GO terms related to mitochondria, lipid storage, triglyceride homeostasis, ATP-binding cassette (ABC) transporter complex, and more. 

Enriched KEGG pathways in the HYD-SBSR group compared with the model group encompass the PPAR signaling pathway, fat digestion and absorption, glycerophospholipid metabolism, and cholesterol metabolism. These pathways are closely associated with the body’s antioxidant status and changes in MDA and LP indicators. PPARs, in particular, play a pivotal role in lipid metabolism, mitochondrial function, and antioxidant defense, helping mitigate oxidative stress [39]. Enhanced expression of ABCG5 and ABCG8 transporters following HYD-SBSR suggests a potential mechanism for reducing oxidative stress by promoting cholesterol excretion and metabolic balance [40,41]. This antioxidant effect is indicated by restored SOD, CAT, and GSH-Px activities and decreased MDA levels, which contributes to the reduction in obesity and hepatic steatosis in the liver. The proteomics analysis provides initial insights into the mechanisms underlying HYD-SBSR’s in vivo actions, paving the way for comprehensive research on its promising applications in oxidative stress-related conditions.

## 5. Conclusions

In this study, we initiated our research by creating an oxidative stress model using H_2_O_2_ on B16F10 cells, aiming to assess the potential cytoprotective and repair effects of HYD-SBSR. The results we obtained pointed toward a decrease in apoptosis rates and an improvement in resistance to oxidative stress upon treatment with HYD-SBSR. Additionally, our findings suggested that HYD-SBSR exhibited significant properties in terms of facilitating cell repair.

By incorporating the outcomes of antioxidant enzyme and peroxide analyses at the cellular level, further investigations were conducted to evaluate the impact of HYD-SBSR on antioxidant enzymes and peroxides in mice. The results exhibited significant antioxidant performance, particularly at lower concentrations of the extract, suggesting a potent ability to enhance antioxidant capacity.

To delve into the underlying mechanisms, we carried out a comprehensive proteomics analysis to identify proteins that were differentially expressed upon HYD-SBSR treatment. Subsequently, GO and KEGG pathway analysis were used to reveal the mechanism pathways associated with HYD-SBSR. Noteworthy pathways included PPAR signaling, fat digestion and absorption, glycerophospholipid metabolism, and cholesterol metabolism. These insights provide a deeper understanding of the cellular and molecular processes that underlie the observed cytoprotective and antioxidant effects of HYD-SBSR.

A comparison of in vitro and in vivo research findings demonstrated a consistent alignment. The inclusion of HYD-SBSR into food products not only addresses sustainability concerns in sea buckthorn production but also has the potential to enhance lipid metabolism within the organism. Furthermore, we speculate that it might even contribute to healthy weight management for individuals.

## Figures and Tables

**Figure 1 foods-12-03322-f001:**
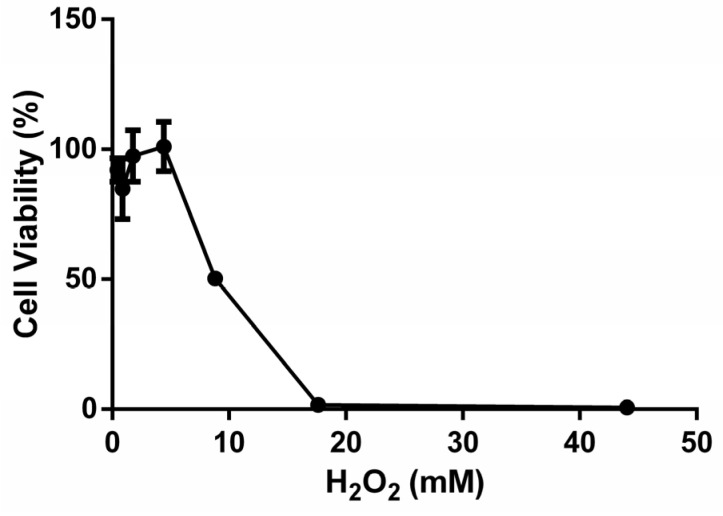
The effect of different H_2_O_2_ concentrations on the cell viability of B16F10. Six parallel wells were used in each group.

**Figure 2 foods-12-03322-f002:**
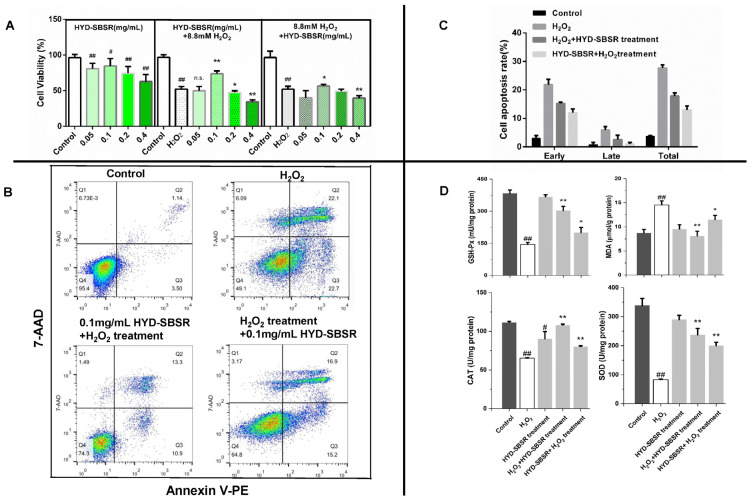
Comparison of the protective and repair effects of HYD-SBSR on the H_2_O_2_-induced B16F10 model. (**A**) Cell viability of B16F10 treated with different concentrations of HYD-SBSR before or after inducing H_2_O_2_ oxidative stress (HYD-SBSR, only treated with HYD-SBSR; HYD-SBSR + 8.8 mM H_2_O_2_, induce 8.8 mM H_2_O_2_ oxidative stress to B16F10 cells pretreated with HYD-SBSR; 8.8 mM H_2_O_2_ + HYD-SBSR, add HYD-SBSR to B16F10 cells pretreated with 8.8 mM). (**B**) Representative figures of flow cytometry in cell apoptosis (model, H_2_O_2_-induced B16F10 model). (**C**) The results of early and late apoptosis and total cell rate. (**D**) Cellular antioxidant-related indexes (GSH-Px, CAT, SOD, and MDA levels). Compared with the control group, # *p* < 0.05, ## *p* < 0.01. Compared with the model group, HYD-SBSR significantly exhibited protective and repair effects on H_2_O_2_-induced oxidative stress, * *p* < 0.05, ** *p* < 0.01. n. s. means no significance.

**Figure 3 foods-12-03322-f003:**
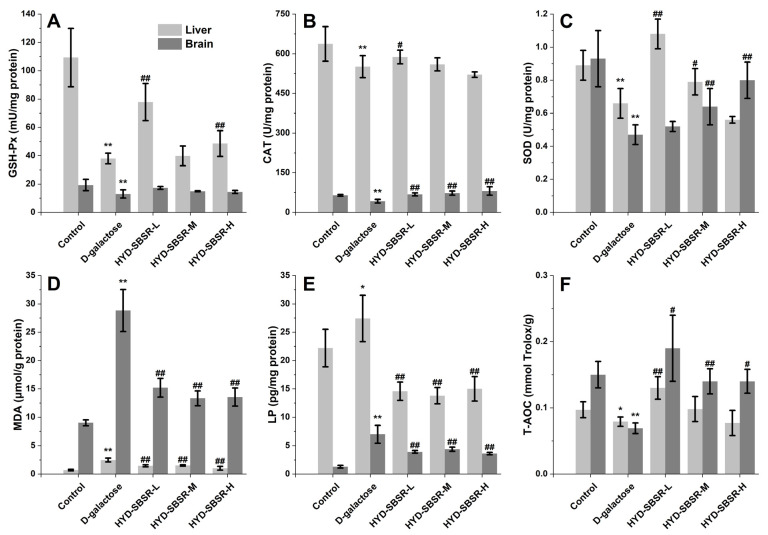
Levels of GSH-Px (**A**), CAT (**B**), SOD (**C**), MDA (**D**), LP (**E**), and T-AOC (**F**) in the liver and brain of five groups of mice (*n* = 10). Control, the control group in which normal mice were only injected with physiological saline every day; D-galactose, the oxidative stress model group only injected with physiological saline every day; HYD-SBSR-L, the oxidative stress model group injected with 100 mg/kg HYD-SBSR every day; HYD-SBSR-M, the oxidative stress model group injected with 300 mg/kg every day HYD-SBSR; HYD-SBSR-H, the oxidative stress model group injected with 600 mg/kg HYD-SBSR every day. Compared with the control group, * indicates a significant difference, *p* < 0.05, and ** indicates a highly significant difference, *p* < 0.01. Compared with the model group, # indicates a significant difference, *p* < 0.05, and ## indicates a highly significant difference, *p* < 0.01.

**Figure 4 foods-12-03322-f004:**
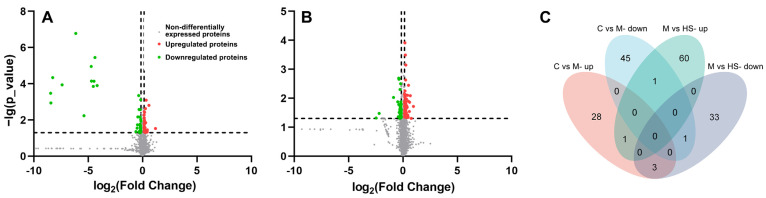
Volcano plots and Venn diagrams of DEPs. (**A**) Volcano plot of the control group vs. the model group, where fold change refers to the ratio of protein abundance in the model group compared to the control group. (**B**) Volcano plot of the model group vs. HYD-SBSR group, where fold change refers to the ratio of protein abundance in the HYD-SBSR group compared to the model group. (**C**) Venn diagrams showing the distribution of overlapping proteins among the control group and the model group (C vs. M- up and C vs. M- down) and the model group and the HYD-SBSR group (M vs. HS- up and M vs. HS- down).

**Figure 5 foods-12-03322-f005:**
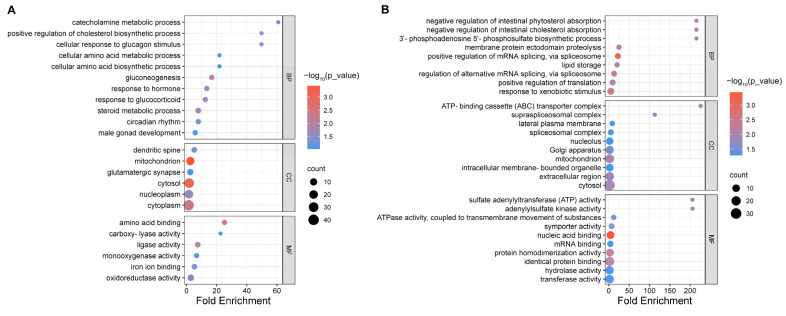
GO enrichment analysis: (**A**) the control vs. the model group and (**B**) the model vs. the HYD-SBSR group.

**Figure 6 foods-12-03322-f006:**
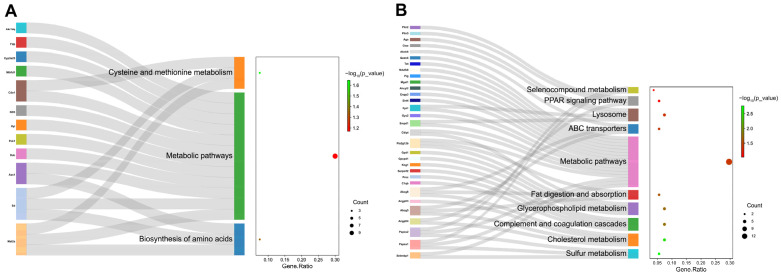
KEGG pathways analysis of DEPs (*p* < 0.05): (**A**) the control vs. the model group and (**B**) the model vs. the HYD-SBSR group.

**Table 1 foods-12-03322-t001:** Overlapping DEPs in the control group vs. the model group and the model group vs. the HYD-SBSR group.

Accession	Symbol	Gene_ID	Control Group vs. Model Group			Model Group vs. HYD-SBSR Group		
			Fold Change	*p*-Value	Regulated Type in the Model Group	Fold Change	*p*-Value	Regulated Type in HYD-SBSR Group
NP_035385.1	Rbp4	19662	1.14615057	0.027117863	up	1.12640839	0.033390243	up
NP_079656.1	Eef1e1	66143	1.163316017	0.03599657	up	0.854203188	0.030938576	down
XP_006518978.1	Farp1	223254	1.255319503	0.000803871	up	0.834798262	0.019648087	down
XP_006509312.2	Aga	11593	1.195095275	0.002335059	up	0.821681249	0.025899789	down
NP_035212.2	Pigr	18703	0.868934766	0.020919521	down	1.198358739	0.0253557	up
XP_017176596.1	Zbed5	71970	0.866735033	0.049608279	down	0.76157783	0.00203081	down

## Data Availability

The data are available from the corresponding author upon reasonable request.

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
