# Peer review of "Ethanolic Extract from Seed Residues of Sea Buckthorn (Hippophae rhamnoides L.) Ameliorates Oxidative Stress Damage and Prevents Apoptosis in Murine Cell and Aging Animal Models"

_foods, 2023, doi:10.3390/foods12173322_

Round 1
Reviewer 1 Report
Comments and Suggestions for Authors
General Comment
The work of Zhongjie Hua 1,2, Jiachan Zhang 1,2, *, Wenjing Cheng 1,2, Changtao Wang 1,2 and Dan Zhao, entitled: “Understanding the role of hydroalcoholic extract from sea buckthorn (Hippophae rhamnoides L.) seed residues in oxidative damaged models in vitro and in vivo”, aimed to study the effects of ethanolic extract from sea buckthorn seed residues (SBSR) on major processes supporting life or organisms and cells, such as redox homeostasis and antioxidant defense system, cell cycle and apoptosis on in vitro murine melanoma model and within mouse animal model of aging mice. Authors aimed to study how extract of SBSR alone affect melanoma cells and how does it act when a model of oxidative stress damage induced by H2O2 is imposed. Smart experimental designed extended to the fact that pre and post treatments were included in regard to order of the treatments with SBSR and H2O2 with experiments on murine cell model. Regarding the animal murine model of aging mice, there were 5 groups, control, L-galactose treated only and low, medium and high SBSR fed of L-galactose treated mice. In total 50 mice. It seems that treatment of control with any of three levels of SBSR was not conducted on animals but it was done on the cell line, which I can understand. Also probably due to very high cost, proteomic experiments were run on liver tissue in a pooled manner, which again I can accept.
All in all, great experimental work, but writing and preparing the manuscript lacks behind substantially. The aspect and results are lovely, but is seems as authors do not praise and like their own work, so to write in accurate, clear, interesting and impactful manner.
Also a question here, how this manuscript aligns with Foods journal scope? Like sea buckthorn has fruit, which is edible, but is the extract of the residues edible? Authors should explain this or give a logical proposal that should fit in the journal scope. Something like when athanoli extract of the seeds is dried it can be encapsultaed and marketed as food supplement. Of course it requires at least in vitro digestion studies.
Since the work on the characterization of the effects of SBS residues started in 2013 as I could notice from this manuscript references and from my own literature search, authors should be objective in the abstract and express so. In the current state of abstract it seems when one reads, that just now the work on SBS residues characterization started and that authors are filling the knowledge gap. The objective situation is that they are deepening the understanding of SBSR effects. This definitely lowers the novelty, but does not mean it necessarily lowers the quality of the manuscript. This manuscript if corrected adequately, has the potential to be one of the favorite in in Section within targeted journal.
However, there is substantial lack of methods description, especially in the supplementary section which is not constrained by the word count and this is something to be objected definitely, including some section in the main text as addressed down below.
Try to use and stress more on the fact that you are dealing with the by-products, e.g. residues due to the nowadays trends of nurturing green and sustainable chemistry and circular economy. If not in the introduction, this point can be placed in worked out in the results and discussion. This is not a must do, this is just and value option that I have seen and if in your shoes, I would use it.
The major weakness of this manuscript is English language style and grammar in a sense that it is not just sole English language but the scientific meaning, accuracy and clarity. Therefore majority of my further specific comments will be addressing this point. It is requested from authors to look for a true knowledgeable and the English language skillful person to polish entire manuscript.
The following comments should be addressed properly.
Specific comment 1
Title. Instead of the present form consider to change it into “Ethanolic extract from seed residues of sea buckthorn (Hippophae rhamnoides L.) ameliorates oxidative stress damage and prevents apoptosis in murine cell and aging animal models“. The English language style of the current version has a great room for improvement, since clarity and formal scientific stile is missing.
Specific comment 2 Abstract
Line 10 – “Hippophae rhamnoides L. has emerged widespread research and application.” Should be written like this (one of correct examples), while keeping the initial idea that aimed to be conveyed (hope I sensed, from the current sentence that it was the authors’ intention):
“Hippophae rhamnoides L. has been widely used in research and application, for almost 2 decades.”
Line 14 – “An H2O2-induced damaged B16F10 model was established to determine the optimal concentration of oxidative damage using the MTT method”…First, it cannot be that scientific model can be damaged or is damaged. Secondly, is it true that authors conducted a research just to find the concentration of H2O2, which will not completely destroyed murine melanoma cells? I regard this more as a tool or technical step and not as a research aim. Authors have lost the true meaning of the sentence or idea that they wanted to convey. Instead, I suggest: “The present study aimed to investigate the ability of ethanolic extract of sea buckthorn’ seeds (EE-SBS) in preventing hydrogen peroxide (H2O2)-induced melanogenesis, cell apoptosis and other aspects of oxidative stress damage (here author can explain exactly what aspects were researched, since I have wrote just what I noticed). This mistaken expression is throughout the manuscript “damaged model” and should be corrected throughout.
Specific comment 3
In the supplementary material the algorithm name is Sequest. Not Request. Also authors did trypsin digestion prior tandem mass spectrometry. So what about the pancreatin? Why it is mentioned here? Maybe it is leftover of some other protocols….like not intended mistake. Pls check. Also many data is missing here, like contamination database has to be mentioned, then to describe main mouse database, then setting in the Protein Discoverer program etc…see some of the papers how it is arranged and you will have clear idea what I meant. Like this, does not look good. For example, consult this paper when modifying the writing of your proteomic material and methods https://www.mdpi.com/2304-8158/11/24/3993
Specific comment 4 Figures
The order of the bars representing experimental groups in Figure 2 is not optimal as well as coloring system. Control first (can be white bars as it suggest no treatment), followed by SBSR treatment. Then it goes control treated with H2O2 and after followed H2O2 treated with SBSR. SBSR alone can be green and H2O2 alone can be white with cross lined, while the treatment that has both will be cross lined with green color as a base. Or choose other color instead of green like orange, but keep logical pattern and order.
Specific comment 5
It is difficult to know which protein is upregulated or down regulated within one pair as shown in the Table 1. Concretely, for Rbp4 is written that it is upregulated. My question is: Is it upregulated in the control or in the model group? When heatmap with protein abundancy via XIC curve area is missing and when no legend is placed on the Figure, a reader cannot know this. This applies through this part of the manuscript.
Specific comment 6.
What kind of treatment was studied in the proteomic part regarding the SBRS group, e.g. HYD-SBSR group? I mean in the material and method description for animals it was said that there were 5 treatments: control, L-galactose treated only and low, medium and high SBSR fed of L-galactose treated mice. So this HYD-SBSR group mentioned under the proteomic liver tissue examination, is it from low or medium or high level of L-galactose treated SBSR? By the way, there are many more uncertainties and non-clarified situations in the manuscript, which is the sign of not thankful and messy writing.
Specific comment 7
Why have you choose value of fold change 1.1 for upregulated and 0.9 for downregulated and you haven’t work with the individual mouse samples but with the pooled sample? At least 2.0 FC for upload and 0,5 for downregulated should be applied. As this is somehow considered a gold standard in the field. Lowering them have to be thoroughly justified but not below 1.5 and 0,75 respectively. Also filtering parameters and statistics description for proteomic quantitation is missing. This section has to be recalculated.
Specific comment 8
Why total identified proteins were used as a basal background database in GO ontologies enrichment analysis? More logical is to use all mouse proteins from the database that was used for protein identification. Like this authors have got skewed and possible less relevant results.
Kind regards
Comments on the Quality of English LanguageThe major weakness of this manuscript is English language style and grammar in a sense that it is not just sole English language but the scientific meaning, accuracy and clarity. Therefore majority of my further specific comments will be addressing this point. It is requested from authors to look for a true knowledgeable and the English language skillful person to polish entire manuscript.
Reviewer 2 Report
Comments and Suggestions for Authors
The study aims to assess the effects of the hydroalcoholic extract of sea buckthorn (Hippophae rhamnoides L.) seed residues (HYD-SBSR) at the cellular and physiological levels, while establishing a correlation between in vitro and in vivo experimental findings.
General comments
-Write the full name of the abbreviations the first time they are cited in the document.
Introduction
Line 65. Add point after the quote.
Lines 89-101: These paragraphs should be in the methodology section.
Methodology
Line 104-113: This paragraph is too large, please divide in several sentences.
Line 114: Should be “B16F10 cells were obtained from the Cell Resource Center
Delete section 2.2. and add the brand and country of origin immediately after the equipment is named in the methodology. For example: “B16F10 cells were cultured in DMEM, supplemented with 10% heat-inactivated fetal 131 calf serum, and a 1% antibiotic-antimycin solution consisting of 100 units/mL penicillin 132 /streptomycin, and 100 U/mL amphotericin at 37 °C in a humidified incubator (brand, place) under 5% 133 CO2.”
Line 255: change ?̅ ± s for mean ± standard deviation.
Section 2.3 is not clear, rewrite it.
Conclusions
Rewrite conclusions according to the objectives of the study.
Comments on the Quality of English Language
Moderate editing of English language required, so it is recommended that the manuscript should reviewer by an native english speaker.
Reviewer 3 Report
Comments and Suggestions for Authors
This presented manuscript by Hua and coauthors, aims to describe the effect of extract obtained from Hippophae rhamnoides L. seed residues after oil recovery on oxidative damage. The study included cell line based in vitro experiments, as well as in vivo with mice. The text is easy to follow and can be potentially interesting for the readers, but it needs to be improved before being accepted for publication. If possible, the answers and comments should be included in a new version of manuscript, since they will allow simpler following of presented data.
Comments:
In the introduction present more information about cell line and mice strain chosen for studies.
Line 89- present the concentration of mixture of water and alcohol used for extraction rather than term “hydroalcoholic”
2.3 - Present more details about the Sea buckthorn seed residues, i.e. variety. Frankly speaking, despite writing the reference, the authors should present more detailed methodology for extract obtaining. What is more, present more detailed characterization of extract components (the results published in reference are from 2018) with the proper identification of constituents and presentation of quantitative data. Without these results all other studies are useless - there are many different varieties of sea buckthorn, and even geological conditions strongly impact their biological activity. The data from 2018 is very general and does not present the proper analysis. Present the concentration of each constituent in dry mass of Sea buckthorn seed residues and in extract added to cells.
What was the diluent used for extract preparation? What was the stock solution used and how was the extract added to the cells (wells)?
2.4- what is the “B16F10” cell line?
Line 140-141 – present the proper volume of each component
Line 142-143 – was DMSO added directly to the cells with medium and MTT?
Present H2O2 concentration in M – this unit is more often used and will allow comparison with other studies. Why in the studies for oxidative damage 300 microg/mL was used? Was it not to strong for checking the extract effect on H2O2 induced damage?
Why the Authors used in the protection and repair studies concentrations of extract equal to 0.05 mg/mL and 0.1 mg/mL, since both of them decreased the cells metabolic activity (determined with MTT assay) by almost 20%? Therefore, in my opinion, “In the study, HYD-SBSR at the concentration of 0.1 mg/mL was selected as the subsequent experimental condition” was wrong and there should be chosen concentration less than 0.05 mg/mL.
2.8 – present the details for determination of GSH-Px, CAT, SOD and MDA in B16F10 cells (there are only kits mentioned), i.e. cells density, the type of plate used for cells seeding, volume of medium and other components, cells treatment, cell lysate preparation used for determination, method of detection. In presented way none of the step can be repeated.
2.9 – present details of cells seeding and experiments preparation, etc. Especially, if the cells were detached for FC after their treatment of before?
2.10 - why D-galactose was used for oxidative stress induction? Why were used doses of HYD-SBSR equal to 100, 300 and 600 mg/ (kg body weight)? What is the name of mice strain used in the experiment?
Line 219 – what liquid was used to obtain 10% homogenate?
All Figures – there need to be added details of experiments in their captures; n of experiments.
Instead of “Model” in text and figures connected with cell line in vitro study use H2O2.
Why 0.05 mg/mL is more cytotoxic than 0.1 mg/mL?
Figure 2D – confirm that concentration of HYD-SBSR used was 0.1 mg/mL.
3.3 – why the Authors did not determined serum levels of GSH-Px, CAT, SOD, MDA, LP and T-AOC?
Instead of “Model” in text and figures connected with in vivo study use D-galactose.
In the discussion there is no comment/correlation of HYD-SBSR composition with observed biological activity, and this point is very important.
Overall, the manuscript requires at least the major revision and repetition of some experiments before its acceptance for publication.
Round 2
Reviewer 1 Report
Comments and Suggestions for Authors
Dear Authors,
thank you for implementing the most of the comments imposed.
I have just one request regarding the Figure 2C. it is quite tiny comapred to 2A,B an D figures. Is it possible to enlarge it? I will not opt now for minor revision, and will choose, accept in the present form, however, plase bear in mind that we write papers for other people to read it. Therefore it would be desirable for this Figure to be larger.
Kind regards